# Replanning in Advance for Instant Delay Recovery in Multi-Agent Applications: Rerouting Trains in a Railway Hub

**Primary Keywords:** *(1) Applications*

## Abstract

Train routing is sensitive to delays that occur in the network. When a train is delayed, it is imperative that a new plan be found quickly, or else other trains may need to be stopped to ensure safety, potentially causing cascading delays. In this paper, we consider this class of multi-agent planning problems, which we call Multi-Agent Execution Delay Replanning. We show that these can be solved by reducing the problem to an any-start-time safe interval planning problem. When an agent has an any-start-time plan, it can react to a delay by simply looking up the precomputed plan for the delayed start time. We identify crucial real-world problem characteristics like the agent's speed, size, and safety envelope, and extend the any-start-time planning to account for them. Experimental results on real-world train networks show that any-start-time plans are compact and can be computed in reasonable time while enabling agents to instantly recover a safe plan.

## Introduction

When a train is delayed, operators must replan its route to minimize the effect on other trains. Minor delays happen frequently and can compound into larger holdups that propagate through the network. Replanning is thus a complex task that must be solved as soon as possible. Yet, the search space grows exponentially in the number of trains to be accounted for, although in practice replanning is still done manually by human operators. Our focus is to swiftly handle holdups that can be resolved by rerouting only the delayed agent. By quickly reacting without infringing on the pre-existing plans of other agents, we are more likely to avoid the cascade of delays. Railway network delay planning is representative of a set of planning problems with similar constraints and agent interactions, including moving automated guided vehicles in container terminals and navigating self-driving cars in dense road networks. We name these Multi-Agent Execution Delay Replanning (MAEDeR) problems. These are single-agent problems that occur during the execution of a multi-agent plan.

We develop a method for solving MAEDeR by replanning in advance using any-start-time safe interval path planning (@SIPP) (Thomas et al. 2023), which allows for instant delay recovery. The search procedure computes optimal plans for all possible starting times ahead of execution, so the agent can select a feasible plan once its start time is known. To prepare for a delay, our approach to MAEDeR precomputes an any-start-time plan for each agent, treating the other agents executing their plans as moving obstacles. So, when an agent is delayed, it can instantly recover a safe plan and execute it without affecting other agents.

So far, any-start-time planning has only been used for single-agent problems in grid-based settings. We show how it can be useful in a multi-agent context, and how to apply it to railway hub planning, a setting closer to real-world problems than point robots on a grid. A railway hub is an area with a train station and shunting yards. For instance, Fig. 1 shows the Enkhuizen hub in the Netherlands with platforms, sidings, and a track connecting to the railroad network. While we show the example of dense infrastructure hubs, which are the most difficult to plan, the same problem representation is also applicable to larger railway networks, and similar ones would work for other MAEDeR problems.

Railway hub planning has several characteristics that are emblematic of real-world applications that can be modeled as a MAEDeR problem. First, the agents in these problems are not point agents, because they have a spatial capacity, and thus, a temporal extent. This also enables agents to occupy several locations simultaneously, such as a 300-meter-long train stretching over two tracks. Because real-world problems rarely have one type of agent, we allow for heterogeneous agents in terms of size and speed. We create a reduction from railway hub planning to an @SIPP graph that inherently encodes the direction of an agent. In a railway hub, switches constrain the possible moves a train can make, which we model with directed edges. Finally, we include context-dependent safety measures that agents must respect, like a variable headway: the time between two consecutive trains, which depends on the relative travel directions.

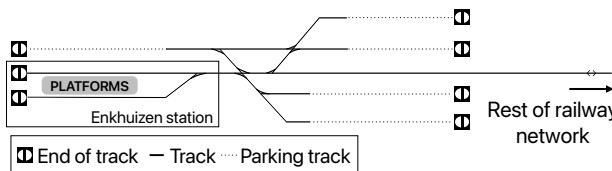

Figure 1: Layout: Enkhuizen railway hub in the Netherlands [Adapted from SporenPlanOnline].

Our main contribution is instantly solving delay response in railway hub planning by applying any-start-time planning to multi-agent execution delay replanning. We recover a safe plan to reroute the delayed train, without affecting other trains' plans. The precomputation of safe plans allows us to rapidly recover the ability to handle a new delay. Our method is extendable to other multi-agent settings. We show promising results for handling delays in real-world problems.

## Example: Railway Hub Planning

We use railway hub planning to instantiate our methods in an intuitive application. A railway hub is an area including a railway station and several surrounding shunting yards where trains can be parked and serviced. The railway hub planning problem considers the routing of trains through the station and between shunting yards, as well as navigating them around other trains in the network. This gives rise to a related MAEDeR problem, of railway hub delay replanning.

Imagine two trains such as those shown in Fig 2, with two station tracks 1 and 2, a parking track $P$, two switches 3 and 4, and a track $E$ connecting the station to the rest of the network. Suppose that the initial multi-agent plan is for train I to depart first, and traverse its route from $I$ to $I'$ without waiting, clearing the shared track right before II uses it to travel from $II$ to $II'$. Because their paths cross over the shared track, if train I is delayed, one of the trains may need to wait in order for both to safely reach their destinations. If train I follows the same plan at this later time, it will conflict with train II. Instead, it should pick a new safe plan that respects train II, such as waiting for train II to clear the shared track, and then finishing its route.

This example follows the Dutch railway operation policies, where trains that are being parked wait for ones serving the timetable. Common causes for delays include too many passengers trying to board, hindrances on the tracks, or a servicing action taking longer than expected. In the following sections, we formalize the intuitions of this example into a planning method to solve this class of problems.

## Background

MAEDeR is a single-agent problem in a multi-agent setting. While the classic multi-agent pathfinding (MAPF) problem constructs non-conflicting plans for a set of agents (Stern 2019), MAEDeR does not come up with the initial multi-agent plan. Instead, it solves the problem of how to quickly recover from one delayed agent while the multi-agent plan is being executed. Delay handling has been previously studied in the context of MAPF, although the focus has been on creating initial plans that are robust to delays (Ma, Kumar, and Koenig 2017; Atzmon et al. 2020a). Where delay recovery has been considered, it has been in the discrete-time context, without singling out a single agent specifically.

Previous studies on delay handling for trains have focused mostly on recovering from disturbances in the complete railway network related to the timetable (Bešinović 2020; Cacchiani et al. 2014). However, railway hub operations differ in that they are more flexible to plan and are planned manually. Some studies have focused on delays solely in

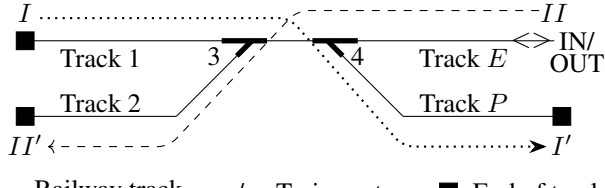

Figure 2: Railway hub planning problem: tracks $(1, 2, E, P)$, switches $(3, 4)$, and trains $(I \rightarrow I', II \rightarrow II')$.

a shunting yard. These consider freight trains in particular (Boysen et al. 2012), or plan robustly to avoid delays (Gardos Reid 2023; van den Broek, Hoogeveen, and van den Akker 2018). Railway hubs have been previously studied as (re)dispatching, including delay response (D'Ariano and Pranzo 2009). However, their method assumes a fixed set of train routes and treats the initial delay handling as a job-shop scheduling problem to reschedule all trains at once. In contrast, our method has two main benefits. First, we do not require preset routes only the origin and destination, allowing the delayed agent more flexibility in adapting to the delay. Second, we resolve the delay without rerouting other agents, so their routes are not changed. In settings where coordinating many agents is non-trivial, the latter is a strong benefit.

## Safe Interval Path Planning (SIPP)

Our method for solving MAEDeR relies on the state space of safe interval path planning (SIPP) (Phillips and Likhachev 2011) and the algorithms for solving any-start-time SIPP (@SIPP) (Thomas et al. 2023). A SIPP problem is a single-agent state-space search problem defined by the tuple $\langle S, E, \delta, s_o, x_g \rangle$. A SIPP search state $\langle x, i \rangle \in S$ has two components: $x$, the configuration (e.g., agent location), and $i = \langle t_s, t_e \rangle$, a *safe interval*, which is a continuous timespan from $t_s$ to $t_e$ when it is safe for the agent to be in configuration $x$. The edges $\langle u, v, i \rangle \in E$ denote an interval $i$ where the agent can safely transition from configuration $u$ to configuration $v$. The cost of an edge is its duration $\delta(u, v)$. The objective of a SIPP agent is to find a minimum duration path to the goal configuration $x_g$, starting from the origin state $s_o$. To solve SIPP problems, Phillips and Likhachev (2011) employ an A* search on this state space, where the objective function $f = g + h$ uses the scalar *earliest arrival time* at a SIPP state as $g$. This returns a single safe optimal plan as its solution, arriving as early as possible at each intermediate SIPP state. Note that several states may share the same configuration, with different non-overlapping safe intervals. Edges between the same pair of states can similarly have several intervals. Because time is continuous, a SIPP search graph is a compact representation of an infinite problem.

For example, take a pedestrian railway crossing. There are two SIPP states: the near side of the crossing and the far side, both of which are always safe. A train passing creates two SIPP edges, the safe interval before the train arrives, and the safe interval after it has passed. In this case, a pedestrian could try and cross at any safe time, but the SIPP graph can represent those infinite actions with only two edges.

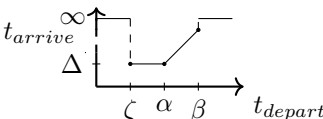

Figure 3: An ATF with parameters $\zeta, \alpha, \beta$, and $\Delta$.

### Any-start-time SIPP (@SIPP)

Recent work on any-start-time SIPP, @SIPP (Thomas et al. 2023) describes how to efficiently generate plans for all start times on SIPP graphs. The augmented SIPP (ASIPP) algorithm described by Thomas et al. (2023) performs a search that is graphically isomorphic to a SIPP search. However, rather than returning a single plan that arrives at a scalar earliest arrival time, ASIPP returns a family of related paths. All paths in the family move through the same sequence of SIPP states but at different times. ASIPP works by 'augmenting' the SIPP search nodes to track earliest *arrival time functions* (ATFs), instead of the scalar $g$ used by SIPP.

The ATF of this path family tells us the earliest arrival time for any departure time along the corresponding sequence of SIPP states. To enable this search, the graph is transformed again, from a SIPP graph to an @SIPP graph. For each edge in the SIPP graph, the source, destination, and edge safe intervals into a simple piecewise linear ATF, defined by the parameters $\langle \zeta, \alpha, \beta, \Delta \rangle$:

$$f[\zeta, \alpha, \beta, \Delta](t) = \begin{cases} \infty & t < \zeta \\ \alpha + \Delta & \zeta \leq t < min(\alpha, \beta) \\ t + \Delta & \alpha \leq t < \beta \\ \infty & \beta \leq t, \end{cases} \quad (1)$$

where $\zeta$ is the earliest time the agent can safely wait at the starting state of the edge, $\alpha$ is the earliest time the agent can safely begin traversing the edge, $\beta$ is the time the edge becomes unsafe, and $\Delta$ is the transit time of the edge. Fig 3 shows an example of an ATF and its parameters.

In a SIPP search, successors are generated with scalar $g$ being the cumulative sum of edge costs along the path so far. In contrast, ASIPP's functional $g$ is the cumulative composition of edge ATFs along the path so far, which maintains the same piecewise-linear structure $\langle \zeta, \alpha, \beta, \Delta \rangle$.

A final algorithm developed by Thomas et al. (2023) to solve @SIPP is called RePEAT, which repeatedly calls ASIPP and restarts a partial expansion A* search with monotonically increasing start times. RePEAT then compiles the plans returned by ASIPP into a set that can be rapidly queried for a plan corresponding to any departure time. Thomas et al. (2023) show that this set of plans produced by RePEAT remains compact in theory and practice.

## Problem Description: MAEDeR

A Multi-Agent Execution Delay Replanning problem (MAEDeR) is defined by the tuple $\langle N, T, \mathcal{C} \rangle$. The infrastructure network $N$ is a set of connected components, which can be represented as a graph with edges between locations. The agents $t \in T$ each navigate through the network. The problem characteristics $\mathcal{C}$ define how the agents interact with the network and each other. These are context-specific and include information to calculate the edge duration $\delta(u, v) : u, v \in N$ and to constrain when edges can be traversed. We refer to $N$ and $T$ collectively as the system, which is safe if all agents have conflict-free plans.

A solution to MAEDeR is a function $\mathcal{F}$ taking an agent $a \in T$ and a positively delayed start time $d$ and returning the shortest safe plan for the delayed agent, or a failure if no feasible plan exists. The returned plan does not require modifications of the plans of any agent other than the one that was delayed. When an agent is delayed, the system is no longer safe until the function returns a new plan for this agent. We refer to this period as the *interval of uncertainty*. The objective of MAEDeR is to provide a solution that minimizes the interval of uncertainty.

### Specifics for Railway Hub Planning

We now illustrate the MAEDeR definition for railway hub planning. In a railway hub, the components of the network $N$ include track segments, platforms, and switches. The trains moving through the hub are the set of agents $T$. The characteristics $\mathcal{C}$ include each train's length $\lambda$ and speed $\nu$, and the length of each track segment $\ell(u, v)$. The duration to traverse an edge can then be calculated as $\delta(u, v) = \frac{\ell(u,v)}{\nu(a)}$.

The constraints for traversing edges in a railway hub relate to the static infrastructure and movements of trains. As trains cannot navigate sharp angles, they have to be reversed to change direction. To do so, the driver has to walk to the other side of the train. Therefore, we need the walking speed of a driver $\omega \in \mathcal{C}$. The constraints that determine when edges can be traversed avoid conflicts and construct a safety envelope for each train, which is a safety buffer between trains. An action's safety envelope is defined by a train's *headway*, which is the time between two consecutive trains. As this is context-specific, we define the *following headway* $\epsilon_f \in \mathcal{C}$ for trains in the same direction and the *crossing headway* $\epsilon_c \in \mathcal{C}$ for the opposite direction. Other MAEDeR problems may share some or all of these characteristics with railway hub planning.

Other planning tasks arise in railway hub planning, like railway freight traffic planning. Freight traffic is done ad hoc depending on the arrival of supply. Since the start and destination are known in advance, we can precompute the any-start-time plan of a freight train. So, our method can be used out of the box for these scenarios, too. Then, the required route can be queried once a freight train is ready to depart.

### When MAEDeR Fails

The MAEDeR objective is to produce a solution function $\mathcal{F}$ that calculates a feasible shortest plan for agent $a$ starting at delayed time $d$. If no safe plan with start time $d$ exists, then $\mathcal{F}$ returns that no such plan exists. How the system reacts to this is problem-specific and outside the scope of this paper. An example reaction could be to execute a multi-agent replanning algorithm. Measures to ensure safety in the meantime are also problem-specific. For example, when planning driverless taxis, this could be an all-stop order halting all taxis until a new safe multi-agent plan is found (Lu 2023).

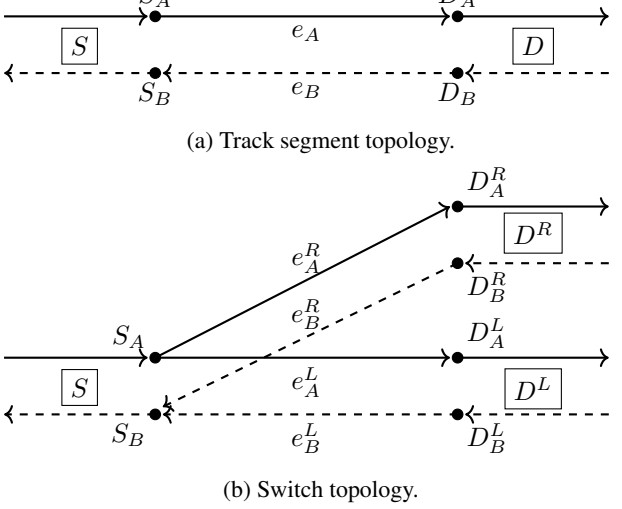

(a) Track segment topology.

(b) Switch topology.

Figure 4: Topology of (a) track segments and (b) switches.

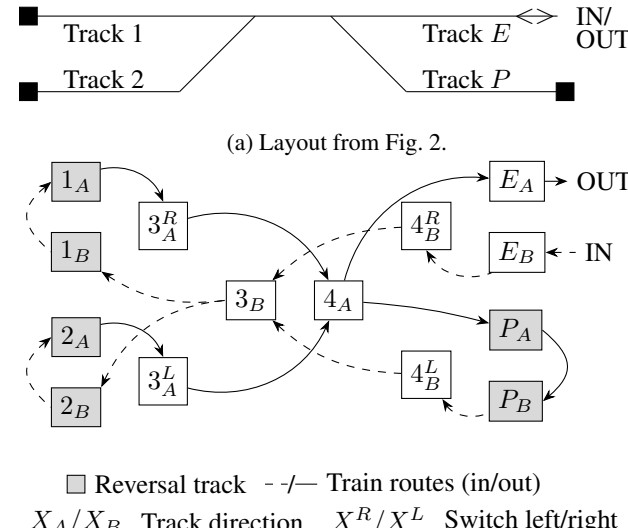

(a) Layout from Fig. 2.

□ Reversal track   - -/— Train routes (in/out)
$X_A/X_B$ Track direction   $X^R/X^L$ Switch left/right

(b) Graph representation of the layout.

Figure 5: Modeling the layout graph (solid versus dashed lines are used to demonstrate the direction of a route).

## Reducing MAEDeR to Any-Start-Time SIPP

To solve MAEDeR problems, we provide a reduction to
@SIPP. In short, for each agent, we transform the network
into a SIPP graph with safe intervals on the states and edges
for that agent, treating the other agents as moving obstacles.
We then compute an any-start-time plan for that agent, so
that when it is delayed, it can instantly recover a safe plan.
From here on, we demonstrate this reduction for the railway
hub planning problem, but this can also be applied to other
MAEDeR contexts, such as navigating automated vehicles
in container terminals or dense road networks.

First, we give an intuition on this reduction. The loca-
tions in the transformed graph loosely correspond to points
in the railway network where track segments meet or join
other infrastructure components. The actual track segments
are represented by edges, one per possible travel direction.
The unsafe intervals result from other trains moving through
the network, blocking safe access to the track(s) that they
occupy. This reduction must obey the physical properties of
the infrastructure, and maintain the safety of the network.

In railway hub planning problems, these properties (char-
acteristics $\mathcal{C}$) are inherent to tracks and switches. Because
trains move forward on the track and cannot simply turn
around, the available edges for a train to follow depend on
the direction it is traveling. For example, in Fig. 4a, two
straight track segments next to each other are connected: $S$
on the left and $D$ on the right. A train can go either left
to right or right to left, based on its initial direction. To al-
low this, we split each location into two co-located states
($A$ and $B$) as is a custom in the railway sector. Now, when
going from the left track to the right track, the train goes
from $S_A$ to $D_A$. The other way around, a train would go
from $D_B$ to $S_B$. We transform the undirected edges of the
train network into pairs of antiparallel directed edges, where
$A$-sides of states have directed edges solely to $A$-sides, and
$B$ only to $B$-sides. The $A$ and $B$-sides are only connected if
this is a track where a train can reverse, for example, at the

end of track 1 in Fig. 5a.

To model a switch, we place three pairs of co-located
states in the graph, as shown in Fig. 4b. Here, we see the con-
junction of three tracks, with $S$ on the left and two tracks $D$
on the right. The two tracks are named $D^L$ and $D^R$ to sig-
nify the left and right sides of a switch, also a custom in
railway operations. A train incoming from track $S$ encoun-
ters the location $S^A$, which has two successors $D_A^L$ and $D_A^R$.
Oppositely, a train coming from either track $D^L$ or $D^R$ will
use the $B$-side and continue its route over point $S^B$. Since a
switch always forms an acute angle between two tracks on
the same side, a train cannot make that turn, so the same-side
nodes are not connected.

These translations are used to construct the graph of a
hub railway layout. An example is shown in Fig. 5. All four
tracks $(1, 2, E, P)$ have two nodes ($A/B$-sides) and switches
have additional $R/L$-nodes on the same side of the switch.
We see that only the reversal tracks $(1, 2, P)$ have their $A$
and $B$-sides connected. Finally, all $A$-sides are connected to
neighboring $A$-sides, and similarly for the $B$-sides.

### Safe Interval Generation

Given the graph of the network infrastructure $N$, we want
to generate safe intervals that allow our agent to navigate
safely given the other agents in the problem. We now show
the interval generation process specifically for railway hubs,
though this can be similarly done for any type of obstacles
moving across an infrastructure. We start by tracing out the
unsafe intervals created by other trains, which are then in-
verted to form safe intervals. Intuitively, a location is unsafe
when it is occupied by (part of) a train, and an edge is un-
safe when a train occupies the start of it. We have co-located
states for each location, and safe intervals are often shared
between them. The antiparallel edges joining a track's two

pairs of co-located states have a more complicated relationship. An edge that is traversed by a train is unsafe until the train has completely departed the edge's origin. On the other hand, the antiparallel edge is unsafe while the train is still occupying any part of the edge.

We calculate the unsafe intervals as follows. Take agent train $a$ traveling from location $u_A$ to $v_A$ beginning at $t_0$. The time to traverse the edge $\delta(u, v)$, and the duration $\delta_a$ of $a$ passing one point on the edge are

$$\delta(u, v) = \frac{\ell(u, v)}{\nu(a)}, \delta_a = \frac{\lambda(a)}{\nu(a)}. \tag{2}$$

The time to fully traverse the edge front to rear is thus $\delta(u, v) + \delta_a$. The unsafe interval for location $u_A$ is

$$i_{u_A} = \langle t_s^{u_A}, t_e^{u_A} \rangle = \langle t_0, t_0 + \delta_a + \epsilon_f \rangle, \tag{3}$$

where train $a$ arrives at $u_A$ at $t_0$, the duration to traverse $u_A$ is $\delta_a$, and we add the headway $\epsilon_f$ to complete the safety envelope. For location $v_A$, the unsafe interval is

$$i_{v_A} = \langle t_s^{v_A}, t_e^{v_A} \rangle = \langle t_0 + \delta(u, v), t_s^{v_A} + \delta_a + \epsilon_f \rangle, \tag{4}$$

where the start time of the interval is the moment the train arrives at location $v_A$ (start time $t_0$ + traversal $\delta(u, v)$), and the end of the interval has the additional duration $\delta_a$ and headway $\epsilon_f$. For the end of the intervals, we need the time for the rear of the train to depart, which is the length of the train over its speed. Co-locations $u_B, v_B$ are unsafe in the intervals

$$i_{u_B} = \langle t_s^{u_B}, t_e^{u_B} \rangle = \langle t_s^{u_A}, t_s^{u_A} + \delta_a + \epsilon_c \rangle, \tag{5}$$

$$i_{v_B} = \langle t_s^{v_B}, t_e^{v_B} \rangle = \langle t_s^{u_B} + \delta(u, v), t_s^{v_B} + \delta_a + \epsilon_c \rangle, \tag{6}$$

so the only difference is the headway $\epsilon_c$ that is included instead of $\epsilon_f$. The edge $e = (u_A, v_A)$ is unsafe from

$$i_e = \langle t_s^e, t_e^e \rangle = \langle t_s^{u_A}, t_e^{u_A} \rangle, \tag{7}$$

which already includes the headway as well. The antiparallel edge $e' = (v_B, u_B)$ has the unsafe interval

$$i_{e'} = \langle t_s^{e'}, t_e^{e'} \rangle = \langle t_s^{u_A}, t_e^{v_A} \rangle. \tag{8}$$

For example, consider the scenario in Fig. 5, where a train II is routed from $E_B$ to $2_A$ (originally shown in Fig. 2). We construct the unsafe intervals for train II, which are shown in Fig. 6. Take train II traveling from location $E_B$ to location $4_B^R$ which takes $\delta(u, v) = 100$ (Eq.2). The train departs $E_B$ at $t_0 = 100$ and its front arrives at location $4_B^R$ at 200 (Eq. 4). The end of the unsafe interval for location $E_B$ is 260, which adds the time for the rear to leave (60) and the headway (100) to the start $t_0 = 100$ (Eq. 3). The end of the unsafe interval for location $4_B^R$ is 360 (Eq. 4). This yields the unsafe intervals $\langle 100, 260 \rangle$ and $\langle 200, 360 \rangle$ for locations $E_B$ and $4_B^R$, respectively. The co-locations $\langle E_A, \langle 100, 210 \rangle$ and $\langle 4_A, \langle 200, 310 \rangle$ have similar interval with only the headway difference $\epsilon_f - \epsilon_c$ (Eq. 5 and 6).

For reversal tracks, the intervals are calculated a bit differently. As mentioned, we must ensure that the train has enough time to be reversed. So, when a train arrives at a dead-end track, there is a buffer time before the internal edge

(a) Information about the scenario for train II ($t$).

| $t_0$ | $\lambda(t)$ | $\nu(t)$ | $\omega$ | $\epsilon_f$ | $\epsilon_c$ |
|---|---|---|---|---|---|
| 100s | 600m | 10 m/s | 1 m/s | 100s | 50s |

☐ Reversal track  - -/— Train routes (in/out)

(b) For each node involved in the move, the associated intervals are given. The four edges used in the movement are shown as thick and their length is given in meters.

Figure 6: Unsafe intervals for train II ($t$) from Fig. 2. These form the obstacles for train I to navigate through.

becomes a safe action. This time is calculated as the time for the driver to walk across, using driver speed $\omega$ and train length $\lambda(a)$. The unsafe interval for such a location $y$ is

$$i_y = \langle t_s^y, t_e^y \rangle = \left\langle t_s^y, t_s^y + \frac{\lambda(a)}{\omega} + \epsilon_f \right\rangle, \tag{9}$$

where both co-locations $y_A, y_B$ of the track get the same interval, as well as the edge $(y_A, y_B)$. For example, the locations $2_A$ and $2_B$ have such an interval in Fig. 6.

For a switch, consider the topology shown in Fig. 4b. Take a train traveling from $S_A$ through the switch to $D_A^R$. The switch should be unsafe to traverse while the train is moving through it, but it should be safe for a train to wait at $D_B^L$ for the switch to clear. The unsafe intervals for $S_A$ and $S_B$ can be calculated using Equations 3 and 5, respectively, and the same for $D_A^R$ (Eq. 4) and $D_B^R$ (Eq. 6). Since state $D_A^L$ is co-located with $D_A^R$ they share the same interval. State $D_B^L$ has no unsafe interval for this move, because the train could technically wait here. For the edges, edge $(S_A, D_A^R)$ has the interval $\langle t_s^{S_A}, t_e^{S_A} \rangle$ (Eq. 7) and edge $(D_B^R, S_B)$ has interval $\langle t_s^{S_A}, t_e^{D_A^R} \rangle$ (Eq. 8). Since there is in practice only one track part that is the switch, the edges all share the same intervals. So, the intervals for $(S_A, D_A^R)$ and $(S_A, D_A^L)$ are the same based on Equation 7 and the intervals for $(S_A, D_A^L)$ and $(S_A, D_A^L)$ are equal based on Equation 8. The example (Fig. 6) shows the resulting intervals for switches 3 and 4.

Following this reduction, we have a SIPP problem with safe intervals on states and edges. We can apply the approach of Thomas et al. (2023) to compile the safe intervals into edge arrival time functions, and then solve it as an any-start-time SIPP problem.

## Solving MAEDeR

The generic planning loop for solving MAEDeR consists of the following points in time we call milestones:

**Unsafe** when a delayed agent learns it is delayed,

**Solve** the MAEDeR function is applied,

**Safe** when the delayed agent regains a safe plan,

**Recompute** a new MAEDeR function is derived,

**Recovered** when the system can handle a new delay.

The interval of uncertainty is between unsafe to safe. An effective method solving MAEDeR minimizes the uncertainty interval. Moreover, it minimizes the time to recompute a new solution, allowing sooner handling of a second delay.

We describe two algorithms for solving MAEDeR: replanning SIPP (rSIPP) runs a new SIPP search for the delayed agent, while any-start-time planning for MAEDeR (@MAEDeR) queries an any-start-time plan. The rSIPP solution consists of a set of SIPP graphs. Each agent has a corresponding SIPP graph with safe intervals where all other agents are treated as moving obstacles. When a delay is encountered, rSIPP selects the SIPP graph of the delayed agent and runs a SIPP search to find a new safe plan starting after the delay. The solving milestone for rSIPP searches the precomputed SIPP graph, while the recomputation milestone precomputes the SIPP graphs.

@MAEDeR trades increased precomputation time for eliminating the interval of uncertainty. The precomputation of @MAEDeR generates the same set of SIPP graphs for each agent as rSIPP. These graphs are then transformed into @SIPP graphs, and we replan in advance using RePEAT to compute the any-start-time plan for each agent as part of the precomputation. The @MAEDeR solution is the set of every agent's any-start-time plan. When an agent is delayed, the corresponding plan is queried at the proper start time. The solving milestone for @MAEDeR queries the any-start-time plan, while the recomputation milestone precomputes the @SIPP graphs and runs the RePEAT searches.

The query operation is a logarithmic-time binary search on a compact collection of plans ordered by their applicable time, and as such is effectively instant. In contrast, even reading the SIPP graph is linear time, and rSIPP or any other method that uses search to recover a safe plan scales in the problem size. @MAEDeR effectively decouples the duration of the interval of uncertainty from problem size, minimizing the duration as much as possible without prior knowledge of the delay. We also highlight that the first step of @MAEDeR's recomputation is to generate the SIPP graphs needed for rSIPP, meaning that @MAEDeR can fall back to rSIPP if another delay happens before it has finished recomputing the any-start-time plans. Additionally, it will reach rSIPP's recovery point before rSIPP would have because its interval of uncertainty is shorter.

| $\lambda(t)$ | $\nu(t)$ | $\omega$ | $\epsilon_f/\epsilon_c$ |
|---|---|---|---|
| 100-2000 | 5-50 | 0.5-5 | 50-500 |

Table 1: Sampled values for scenario generation $\forall t \in T$.

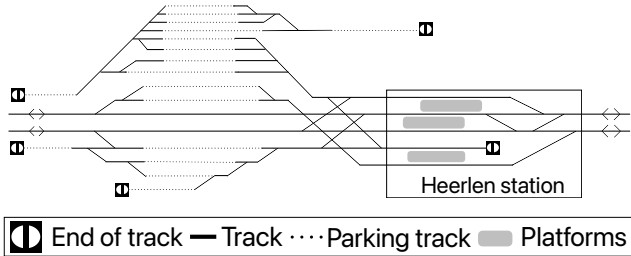

Figure 7: Layout of the Heerlen Railway Hub.

## Experimental Evaluation

The goal of our experiments is to empirically evaluate the performance of rSIPP and @MAEDeR, including their relative performance and our claim that @MAEDeR is 'effectively instant'. We want to answer the following question:

**Q1** Is the interval of uncertainty for @MAEDeR significantly shorter than for rSIPP?

We also demonstrate the appeal of both methods in practice:

**Q2** Are the recomputtation times for both methods using realistic scenarios reasonable for practical application?

### Data

We created our own dataset based on real-world data from two Dutch railway hubs. The smaller one is based on the station of *Enkhuizen* (Fig. 1). The larger one is the station of *Heerlen* shown in Fig. 7), which also has 'free tracks' that can be entered from both sides. The hub layouts comprising the network $N$ were computed manually based on the information available online [1] , taking the length of tracks in meters. We assign all platforms and parking tracks to be places where trains can turn around. Intermediate track segments or ongoing tracks (like IN/OUT in Fig. 5) do not allow trains to reverse. This way, trains are not allowed to stop in the middle of a track where other trains can still be traveling.

For the Enkhuizen hub, we created three scenarios with a different total number of trains. The small scenario (6 trains) was constructed manually and the medium scenario (13 trains) is based on the actual timetables showing the necessary moves on a Tuesday morning (October 31, 2023).[2] This scenario uses realistic headway times (Liu and Han 2017; Wang, Liu, and Zeng 2017), train speeds, and train lengths. Finally, we generated a large scenario of 25 trains (more is unrealistic as the Enkhuizen hub does not have enough tracks for that many trains). For the Heerlen hub, we also generated scenarios with 6, 13, and 25 trains for comparison. Additionally, as this layout is much bigger, we also created a scenario with 50 trains. For each scenario, we created different instances by assigning a different train as our agent, so we had variations of the same scenario.

The scenario generation samples several values using a given random seed (see Table 1). Each train gets a set of routes, which define start and end locations, and there can be

---

[1] sporenplan.nl, openrailwaymap.org

[2] ns.nl/reisplanner, treinposities.nl, treinenweb.nl/materieel

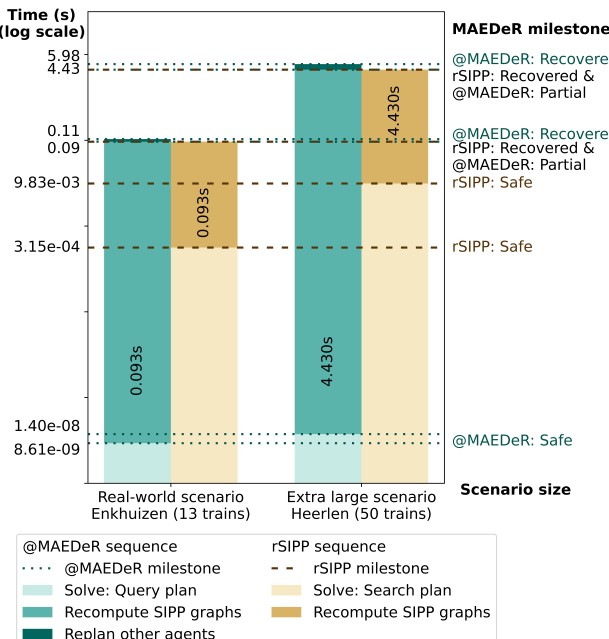

Figure 8: Comparing the average time to reach the milestones described in Section Solving MAEDeR.

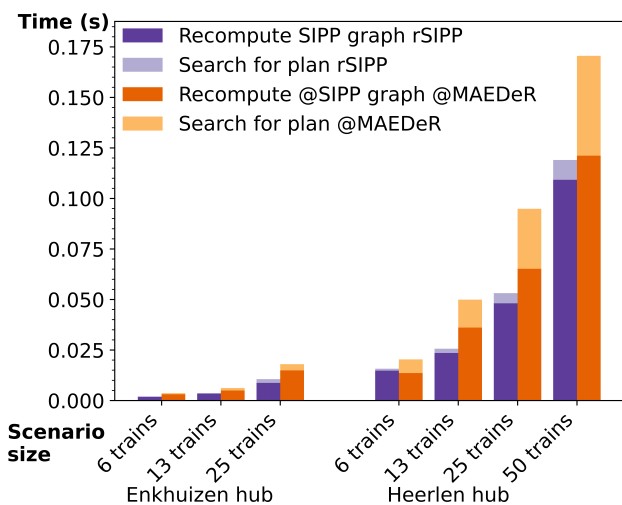

Figure 9: Average replanning runtime of the algorithm for different size scenarios on the Enkhuizen layout.

either 1, 2, or 3 ordered subgoals for the train to reach. The start time of a route is sampled from an interval of $\langle 0, 1000 \rangle$. We find the shortest path for each route, and the end time of the route is calculated using the agent's sampled speed. Successive trains are generated with progressively increasing start times. If the trains in the scenario have no conflicts, then this route is included in the scenario and we move on to the next train route. Otherwise, a new route is sampled until we have the number of required routes for the total number of trains.

## Implementation

The code to replicate our experiments is available on GitHub.[3] We ran our experiments using an Apple M1 Pro processor. The **Q1** and **Q2** search results use an implementation of rSIPP and RePEAT in an efficient C++ code base, using the same search code and data structures when possible. The **Q2** graph computation results currently use an implementation of SIPP and @SIPP graph generation in Python.

## Results and Discussion

Fig. 8 compares the two algorithms in their time to reach the milestones described in the Solving MAEDeR section. To answer **Q1**, we measured the interval of uncertainty, which is the time to reach the **Safe** milestone. In Fig. 8, we can clearly see a significant difference between the algorithms, @MAEDeR's interval of uncertainty is tens of nanoseconds, while rSIPP takes almost 10 milliseconds for the larger scenario, which is 100,000 times slower.

---
[3]Withheld pending acceptance to preserve anonymity.

This result provides empirical support for our assertion that looking up an any-start-time plan is effectively instant. Theorem 1 shows that it is impossible to construct a method that involves communicating between multiple agents or a central planner with a shorter interval of uncertainty than @MAEDeR. Furthermore, the interval of uncertainty is over before the other trains ever knew it began.

**Theorem 1.** *In MAEDeR problems of the scale of railway hub planning, @MAEDeR ends the interval of uncertainty before any other agent can be informed of the delay.*

*Proof.* Our empirical results show that @MAEDeR ends the interval of uncertainty within tens of nanoseconds. Trains within train hubs are generally separated by more than tens of meters. The speed of light is $\approx$0.3 m/ns, and information cannot travel faster than the speed of light. Thus, the delayed agent has ended the interval of uncertainty before any other agent could physically receive word of its delay. □

For **Q2**, we compare the milestones of both methods for different scenario sizes in Fig. 9. These scenario sizes are representative of real-world scenarios for these hubs, both methods offer attractive runtimes to recompute the plans online to allow a new delay to be handled. These attractive runtimes would only improve with a more performant graph generation implementation. This would also increase the separation between @MAEDeR partial and rSIPP recovered. In Fig. 8, the 'recompute SIPP graphs' bars are the same size for each scenario. So, @MAEDeR is ready to use rSIPP ~1 ms before rSIPP is.

Finally, we consider several additional benefits of the @MAEDeR solution. Besides handling delayed trains, human operators could benefit from seeing safe intervals to plan ad hoc freight traffic or being shown the ATF for any train path in the railway hub. Train paths are the regular routes that trains can make, so for any new train that has to be planned ad hoc, we can quickly look up when this train

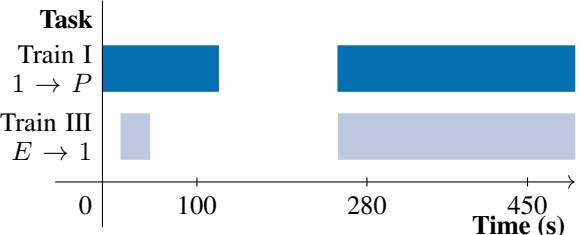

Figure 10: Safe intervals for train I from Fig. 2 and train III.

can safely make its route. In Fig. 10, Train I's intervals show when it can move around train II, and the new train III's intervals give its possible plans around train I and II from Fig. 2. Train III represents a common scenario in real-world railway hub planning, especially when considering freight traffic. These trains are often dependent on the arrival time of a container ship, so they need to be planned last-minute.

## Related Work

MAPF with Delay Response considers the delay of agents by predicting these upfront and handling them during plan execution (Ma, Kumar, and Koenig 2017). The effectiveness of this approach obviously depends on the accuracy of the predictions. A general framework for MAPF that is more robust against delays is called $k$-robust MAPF (Atzmon et al. 2020a) and extends the safety envelope for a grid node with $k$ time steps. However, a delay suffered by one agent is often propagated to other agents, something we wish to avoid. One solution to this uses a temporal network for post-processing, but delays are still handled ad hoc (Hönig et al. 2016). This can be a complex task to solve and we wish to avoid lengthy computations while the system is in an unsafe state, so a precomputed recovery plan is safer.

There has been much work on speeding up trajectory planning in dynamic environments (Nantabut 2023, inter alia). However, we wish to avoid planning from scratch when a delay is encountered. MAPF with heterogeneous agents has been proposed before (Atzmon et al. 2020b) and a MAPF model was developed specifically for train routing (Švancara and Barták 2022), which also includes the length of a train agent, but does not allow for different agent speeds. However, their focus is on allocating tracks for the timetable, not a railway hub, and they do not include the dynamics of the environment. All the MAPF works cited here handle delays by either predicting them, planning such that they are less likely to occur, or planning ad-hoc in response, while the main benefit of our method is precomputing new plans for possible delays.

MAEDeR and any-start-time planning can be seen as precomputing a policy for any delay, or similarly as a type of universal plan (Schoppers 1987). Universal planning is "an (almost) universally bad idea" (Ginsberg 1989) because universal plans generally grow exponentially with problem size. However, @SIPP is one of the exceptions: plans grow linearly with problem size (Foschini, Hershberger, and Suri 2014; Thomas et al. 2023).

## Conclusion

This paper provides a solution for handling delayed trains in a railway hub. When an agent is delayed, it can instantly recover a safe plan and resume execution, eliminating the interval of uncertainty that the railway hub system is unsafe. More generally, we demonstrated how to apply any-start-time planning to a multi-agent real-world setting. We defined the multi-agent execution delay replanning problem (MAEDeR) and showed how to transform such practical problems into safe interval path planning (SIPP) formulations. These can then be used in an any-start-time planning approach to recover safe plans for all agents based on unknown delay. Because these plans can be computed prior to execution, the agent can immediately use its safe plan once its start time is known.

Compared to earlier work on any-start-time SIPP, our approach extends beyond the grid-based pathfinding domain. Moreover, we allow for different agent sizes and speeds, we let the agent spatially occupy several locations, and we inherently encode the movement direction in our graph. The latter is especially important in MAEDeR problems where the agents cannot easily turn around, so we ensure that the computed path is always feasible. Our method can also deal with different types of safety envelopes, that go beyond conflict risks and can depend on the travel direction.

The experiments showed that the lookup time for a safe plan is instantaneous, so agents can recover their safe plan and immediately execute it without extra waiting time. Furthermore, we show that the use of a SIPP graph with safe intervals still allows for fast replanning, enabling other agents to quickly react to one delayed train. Our method scales well, solving real-world size scenarios in a reasonable time, allowing operators to respond rapidly.

Any-start-time planning provides the benefit of precomputing arrival times for different scenarios, so the applicable one can be chosen at execution time. In the railway hub planning domain, a train that arrives late can cause propagating delays through the railway network, resulting in unknown arrival times. With this method, the delay propagation can be minimized by allowing trains to recover their safe plans and continue their execution immediately upon their arrival. When making the initial schedule, we can also ensure that good departure time alternatives are available for each train in case it is delayed. Moreover, the approach can also be used for the ad hoc planning of new trains, which is very common for freight traffic. Our approach shows that any-start-time planning is useful in train routing and can be applied to other MAEDeR problems, like moving automated guided vehicles in a container terminal or navigating self-driving cars in dense urban areas.

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
