# OpenReview forum: "Replanning in Advance for Instant Delay Recovery in Multi-Agent Applications: Rerouting Trains in a Railway Hub"
_icaps-conference.org/ICAPS/2024/Conference — ICAPS 2024_

### Official Review · Reviewer_LDNH · 2024-01-14

**Significance And Importance:** 1
**Soundness:** 3
**Novelty:** 2
**Clarity:** 3
**Overall Evaluation:** 1
**Confidence:** 3

**Weaknesses:**

0: Minor weaknesses requiring some work to be addressed for the paper to be accepted.

**Contributions Of The Paper:**

The paper proposes a new method for solving MAEDeR problems in railway hubs, called @MAEDeR. This method is based on any-start-time planning and has two main advantages over existing methods: it can significantly reduce the interval of uncertainty and it has a shorter recomputation time.

**Ethical Considerations:**

(5) Excellent: The paper comprehensively addresses all of the applicable ethical considerations

**Nomination For Best Paper:**

No

**Questions For Authors:**

1. Does the approach consider the transfer of trains by some users?
2. From the point of view of a Multi-Agent Plan Execution, it seems to be reparing the MAP instead of replanning the MAP (generate a new plan from scratch).

**Reproducibility:**

4: Authors promise to release code and domains (whichever apply).

**Strengths Of The Paper:**

The paper evaluates @MAEDeR on a dataset of real-world scenarios and compares it to a baseline method called rSIPP. The results show that @MAEDeR consistently outperforms rSIPP in terms of both the interval of uncertainty and the recomputation time.

The paper also discusses some limitations of the study, such as the fact that it only considers delays in the agent's start time and does not consider delays in the other agents' plans. The authors suggest that this could be a topic for future research.

**Weaknesses Of The Paper:**

The paper could:

- provide more details on the algorithm, which is used by @MAEDeR to compute any-start-time plans.
- discuss how the methods can be extended to handle other types of delay scenarios.
- discuss the trade-offs between @MAEDeR and rSIPP, such as the amount of precomputation time required.

Minor:
- provide a clear definition of what is an Agent for the authors.

---

> ### Author Rebuttal · Authors · 2024-01-26
>
> 1. No, we do not take into account the fact that some passengers may be planning to transfer from one train to another at a station.  We are already achieving the fastest possible new plan for a delayed train, and delaying another train is against standard procedures, as it is unfair to other passengers and can result in cascading delays.  We will be more clear about this in the description of railway hub planning.
> 2. Yes, since we are replanning a single agent, this can be viewed as a repair to the global multi-agent plan. In our context, we felt replanning is a more appropriate term, although we can add a note in Related Work on how this relates to repair in multi-agent plan execution.
>
> Additionally, we will clarify our usage of the term agent.
>
> Finally, we will include a more explicit reference to the paper implementing the any-start-time search and explicitly mention the difference in computation time between @MAEDeR and rSIPP.

---

### Official Review · Reviewer_zNWL · 2024-01-22

**Significance And Importance:** 2
**Soundness:** 3
**Novelty:** 2
**Clarity:** 3
**Overall Evaluation:** 1
**Confidence:** 3

**Weaknesses:**

2: No major or minor weaknesses.

**Contributions Of The Paper:**

This paper addresses the challenge of efficiently replanning trains at a railway hub in the face of uncertainties related to the origin dispatch. Delays frequently occur at the origin, necessitating the recomputation of plans. The approach taken in this study involves addressing one delayed train at a time, treating all other trains as dynamic obstacles with accurately known plans. This perspective is applied to formulate the problem as a graph planning problem. During the planning process for the delayed train, safe intervals on the graph's vertices and edges are derived from the plans of the other trains. The paper introduces the application of the Any-start-time SIPP algorithm, as presented in a recent paper, to generate optimal plans for all possible start times of the delayed train. The paper proposes to recover an optimal plan almost instantaneously once the delay at the origin is known. A key technical contribution lies in demonstrating how hub networks can be effectively modeled as graphs, and safe intervals can be established based on the plans of other trains. The paper concludes by comparing the implementations of the Any-start-time SIPP algorithm and the standard SIPP in terms of the time required to recover an optimal plan and the duration each planner would leave the system in an unsafe state.

**Ethical Considerations:**

(1) Not Applicable: The paper does not have any ethical considerations to address

**Nomination For Best Paper:**

No

**Questions For Authors:**

1. Pg 2, Line 132: How do Boysen et al. 2012, Reid 2023; van den Broek et al. model the problem? A few sentences about their respective modeling approaches would be helpful to the general reader.
2. Pg 2, Line 143: The second point contrasting your method with others is factually incorrect with respect to the job shop approach of D’Ariano and Pranzo 2009 is incorrect. They too do not change the routes of the other trains, they only resequence the order in which trains use each segment. So I would consider rephrasing that sentence to reflect this exception.
3. Pg 3, what is the complexity of Repeat SIPP in terms of the graph size and #safe interval and #start times?
What is the time complexity for retrieving an entire plan given a start time?
4. Pg 4, How general is the transformation procedure described for transforming the network into a SIPP graph? Does it work any network topology? For example, say we have an undirected graph with nodes P, Q, R, S and T, and edges (P, R), (Q, R), (R, S) and (R, T). Let us assume that a train entering R from P can transition to S or T, while a train entering R from Q can only transition to S. Is the procedure described capable of modeling these scenarios?
5. Pg 6, How many track segments are there in Enkhuizen and Heerlen? This information can be helpful to understand the scalability of the algorithms with the graph size.
6. Pg 6, In lines 456-461: it says the complexity of retrieval for MAEDeR is logarithmic, but logarithmic with respect to what?
7. Pg 7, In line 514, it says “We find the shortest path for each route”. What is the distinction between path and route in this context?
8, Pg 7, In Fig 8, what happens under “replan other agents”?
9. Pg 7, In line 564, it says recompute SIPP graphs are the same in each scenario referring to Figure 8, but clearly it isn’t (I’m comparing light teal with light brown\golden). What am I missing?

**Reproducibility:**

4: Authors promise to release code and domains (whichever apply).

**Strengths Of The Paper:**

The paper introduces a compelling real-world application of the Any-start-time SIPP. Train scheduling, being a dynamic system, requires simultaneous replanning during execution. Consequently, the safety certification of the system is not always guaranteed during the replanning phase. While I don't believe the system proposed in the paper has the potential to replace existing solution approaches, it does offer a valuable tool for training dispatch engineers to quickly handle origin delays.

**Weaknesses Of The Paper:**

In highlighting these shortcomings, it's important to note that I lack prior experience in railway scheduling. Consequently,  some of my criticisms can be inaccurate.
1. When plans are generated using Any-start-time SIPP, the duration of their practical validity raises concerns. The planner's reliance on flawless execution by other trains in temporal space, as assumed by safe intervals, is seldom realized in practice. Consequently, the pre-computed plans by Any-start-time SIPP may prove infeasible when the origin dispatch is realized, raising questions about the method's practical utility. In such situations, opting for SIPP might be a more reliable alternative.
2. Speculating, I ponder whether rerouting a delayed train could trigger operational challenges, such as increased setup times for signaling and switching. If this speculation holds true, maintaining the train's route unchanged, akin to the Blocking Job shop approach of D’Ariano and Pranzo in 2009, might be a preferable strategy. The paper can benefit if authors have some real world experience with train scheduling and are able to include operational level flexibilities and constraints.
3. Despite Any-start-time SIPP being adopted from another paper, the paper lacks details on the complexity of plan generation and retrieval concerning graph size, #safe intervals. Instead, imprecise terms like "compact" and "binary" are employed, necessitating a more precise description in the paper. Please refer to my questions for additional context.

---

> ### Author Rebuttal · Authors · 2024-01-26
>
> Weaknesses:
> 1. Trains are required to maintain headway as a safety buffer, allowing them to recover from small perturbations. SIPP and Any-start-time SIPP both rely on the same interval formalism to handle this.
> 2. As the infrastructure allows only a limited number of paths, new plans often use the same paths as before. We will include some statistics on this. Note that the approach of d'Ariano can result in canceled/rerouted trains, so their method is not necessarily preferable operationally.
> 3. We will clarify the final version of the paper, as also noted in the answers below.
> Questions:
> 1. Good idea, we will include brief summaries.
> 2. We will rephrase to be clearer, but note the ROMA system (D'Ariano & Pranzo) does have the ability to introduce new train routes/cancel a train if no conflict-free schedule for the given train routes can be found (p70).
> 3. Foschini, Hershberger, and Suri (2014) find that plans grow linearly with the # of safe intervals and linearly with start time. This part will be moved from related work (615) to background, and clarified.  We will clarify that plan lookup is logarithmic in the size of the any-start-time plan.
> 4. Because of the physical nature of tracks, only a few types of switches are possible, so this scenario is not possible for trains. In a different context, this could be adapted from a four-way switch.
> 5. We will add that Enkhuizen has 8 tracks and Heerlen 29.
> 6. We will clarify: logarithmic in the number of non-dominated segments in the compound ATF, which is upper bounded by the number of safe intervals in the problem.
> 7. A route is a time-independent path, a sequence of locations rather than states.  We will clarify.
> 8. MAEDeR replans in advance, so the 'replan other agents' is the time spent recomputing any-start-time plans for all the trains.  We will clarify.
> 9. It is true, but not visually clear in Fig8 due to the log scale. We will improve the explanation. Note that the lower edges are at 10^-8s and 10^-4s, which is a very small absolute difference compared to the tops at ~10^-1s. The rectangles represent the same values, but due to the log scale, there is a large visual difference at the bottom and no perceptible difference at the top. As labeled on the plot, both are identical at 0.093s/4.43s respectively for the two scenarios. We presented Fig8 using a log scale because linear would make it impossible to see the MAEDeR query time and rSIPP search time.

---

### Official Review · Reviewer_GKrV · 2024-01-22

**Significance And Importance:** 2
**Soundness:** 3
**Novelty:** 3
**Clarity:** 4
**Overall Evaluation:** 2
**Confidence:** 3

**Weaknesses:**

1: Minor weaknesses that are easily fixable.

**Contributions Of The Paper:**

The paper focuses on addressing the problem of replanning in railways to respond to delays quickly to mitigate any cascading impact. The paper focuses on railway hubs, providing a solution that can make improvements on earlier any-start-time SIPP. Experimental results demonstrate the approach can safely recover without incurring extra waiting time.

**Ethical Considerations:**

(5) Excellent: The paper comprehensively addresses all of the applicable ethical considerations

**Nomination For Best Paper:**

No

**Questions For Authors:**

-Are there any other influencing factors that affect performance beyond the number of trains? For example, the size of the section of the railway being controlled? How would someone make decisions over how big of an area to consider as a hub?

**Reproducibility:**

4: Authors promise to release code and domains (whichever apply).

**Strengths Of The Paper:**

-New approach to tackle widespread railway rescheduling challenges to avoid cascading delays. The approach is clearly presented and explained.
-Real-world evaluation demonstrates the suitability of the approach.
-Clear application focus that could allow the approach to be repurposed for other vehicle control where safety is imperative – e.g., air.

**Weaknesses Of The Paper:**

-Although the approach appears to be efficient (Fig 9), it is noticeable that there is at best a linear relationship between the scenario size (number of trains).  This could be problematic when the technique is handling larger problem instances.

---

> ### Author Rebuttal · Authors · 2024-01-26
>
> With respect to the scaling of our approach with problem size:
> The size of the search space is proportional to the number of safe intervals. So as you point out, we'd expect the runtimes to scale linearly with the problem size. We will add a brief theoretical analysis of the runtime of our search procedure. Adding an additional location adds an additional safe interval. Adding an additional train replaces a safe interval with two smaller ones in each safe interval it passes through. In the computation, the recomputation phase is currently the heaviest.
>
> The size of one hub is decided by the railway companies, as this is an area that is controlled by one operation center. Our algorithm can easily handle the largest hubs that are currently in use, although it would be interesting to investigate how parallel processing could be used to gain ever further scalability, should that be necessary.

---

### Meta-Review · Area_Chair_mN7K · 2024-02-05

**Recommendation:** Accept (Oral)
**Confidence:** 5

**Metareview:**

This paper introduces a new technique for efficiently replanning trains at a railway hub in the face of uncertainties related to the origin dispatch. Applying a previously introduced “any-start-time” Safe Interval Path Planning (SIPP) algorithm and utilizing a novel representation of a rail hub network as a graph with safe intervals derived from the plans of other trains associated with relevant vertices and edges, the proposed approach generates optimal plans for all possible start times of a potentially delayed train and then, once the origin delay of the train is known, a new optimal (and safe) plan for the train can be recovered near instantaneously. Experimental results on real-world train networks demonstrate the viability of the approach.

The paper presents a compelling real-world application of the recently introduced “any-start-time SIPP algorithm to the problem of replanning in railways to respond quickly and mitigate any cascading impact. The approach is clearly presented and explained. Its effectiveness is evaluated experimentally on real-world rail hub networks, and the approach appears applicable to other forms of vehicle control (e.g., aircraft) where safety is imperative.

The reviews provide several comments and suggestions for improving the paper that should be considered to the extent possible when producing the final version of the paper (including all revisions that the author(s) have already proposed in their rebuttal comments).

**Ethical Considerations:**

(1) Not Applicable: The paper does not have any ethical considerations to address